# Melanoma Brain Metastases Patient-Derived Organoids: An In Vitro Platform for Drug Screening

**DOI:** 10.3390/pharmaceutics16081042

**Published:** 2024-08-05

**Authors:** Saif-Eldin Abedellatif, Racha Hosni, Andreas Waha, Gerrit H. Gielen, Mohammed Banat, Motaz Hamed, Erdem Güresir, Anne Fröhlich, Judith Sirokay, Anna-Lena Wulf, Glen Kristiansen, Torsten Pietsch, Hartmut Vatter, Michael Hölzel, Matthias Schneider, Marieta Ioana Toma

**Affiliations:** 1Institute of Pathology, University Hospital Bonn, 53127 Bonn, Germany; racha.hosni@ukbonn.de (R.H.); annalena.wulf@ukbonn.de (A.-L.W.); glen.kristiansen@ukbonn.de (G.K.); 2Institute for Neuropathology, University Hospital Bonn, 53127 Bonn, Germany; andreas.waha@ukbonn.de (A.W.); gerrit.gielen@ukbonn.de (G.H.G.); torsten.pietsch@ukbonn.de (T.P.); 3Department of Neurosurgery, University Hospital Bonn, 53127 Bonn, Germany; mohammed.banat@ukbonn.de (M.B.); motaz.hamed@ukbonn.de (M.H.); erdem.gueresir@medizin.uni-leipzig.de (E.G.); hartmut.vatter@ukbonn.de (H.V.); matthias.schneider@ukbonn.de (M.S.); 4Department of Dermatology, University Hospital Bonn, 53127 Bonn, Germany; anne.froehlich@ukbonn.de (A.F.); judith.sirokay@ukbonn.de (J.S.); 5Institute of Experimental Oncology, University Hospital Bonn, 53127 Bonn, Germany; michael.hoelzel@ukbonn.de

**Keywords:** melanoma, brain metastases, *BRAF*, organoids

## Abstract

Background and aims: Brain metastases are prevalent in the late stages of malignant melanoma. Multimodal therapy remains challenging. Patient-derived organoids (PDOs) represent a valuable pre-clinical model, faithfully recapitulating key aspects of the original tumor, including the heterogeneity and the mutational status. This study aimed to establish PDOs from melanoma brain metastases (MBM-PDOs) and to test the feasibility of using them as a model for in vitro targeted-therapy drug testing. Methods: Surgical resection samples from eight patients with melanoma brain metastases were used to establish MBM-PDOs. The samples were enzymatically dissociated followed by seeding into low-attachment plates to generate floating organoids. The MBM-PDOs were characterized genetically, histologically, and immunohistologically and compared with the parental tissue. The MBM-PDO cultures were exposed to dabrafenib (*BRAF* inhibitor) and trametinib (*MEK* inhibitor) followed by a cell viability assessment. Results: Seven out of eight cases were successfully cultivated, maintaining the histological, immunohistological phenotype, and the mutational status of the parental tumors. Five out of seven cases harbored *BRAF* V600E mutations and were responsive to *BRAF* and *MEK* inhibitors in vitro. Two out of seven cases were *BRAF* wild type: one case harboring an *NRAS* mutation and the other harboring a *KIT* mutation, and both were resistant to *BRAF* and *MEK* inhibitor therapy. Conclusions: We successfully established PDOs from melanoma brain metastases surgical specimens, which exhibited a consistent histological and mutational profile with the parental tissue. Using FDA-approved *BRAF* and *MEK* inhibitors, our data demonstrate the feasibility of employing MBM-PDOs for targeted-therapy in vitro testing.

## 1. Importance of This Study

The incidence of malignant melanoma is rising worldwide. Although the five-year survival rate for early-stage melanoma is over 90%, in advanced metastatic stages it decreases to 10%. About 60% of patients with melanoma develop brain metastases; however, those patients are excluded from clinical trials, which hinders the development of new therapeutic approaches for this group of patients. Additionally, the biology of melanoma brain metastases is poorly understood due to the lack of representative research models; hence, there is a critical need to develop novel faithful models. In this study, melanoma brain metastases’ patient-derived organoid culture lines were established from surgical specimens. The organoid cultures faithfully recapitulated the histological and mutational profile of the parental tissue. Drug-sensitivity experiments using the FDA-approved *BRAF* and *MEK* inhibitors, dabrafenib and trametinib, demonstrated the feasibility of employing melanoma brain metastases patient-derived organoids as models for targeted-therapy in vitro testing and for the discovery of novel therapeutic targets. 

## 2. Introduction

Brain metastases are one of the most common and challenging neurologic complications of cancer. It is estimated that approximately 20% of individuals diagnosed with cancer develop brain metastases [1]. Among these metastatic brain lesions, melanoma stands as the third most frequent cause, accounting for 6–11% of all cases after lung cancer (41%) and breast cancer (19%) [2,3]. Up to 60% of patients with melanoma develop brain metastasis during disease progression. The management of advanced-stage cancer, especially after the development of brain metastases, necessitates a comprehensive and multimodal treatment approach, including surgery, radiotherapy, chemotherapy, immunotherapy, and targeted therapy based on the mutational status [1]. *BRAF* is the gene most commonly affected by point mutations in cutaneous melanoma, which in turn leads to the constitutive activation of the MAPK pathway [4]. This promotes tumor progression, but, on the other hand, it represents a molecular target for therapy using *MAPK* inhibitors, such as *BRAF* and *MEK* inhibitors, which have shown response rates of up to 76% in patients with melanomas harboring *BRAF* mutations [5,6]. The development and the integration of immune checkpoint inhibitors, specifically PD-1 inhibitors (nivolumab and pembrolizumab) and the CTLA-4 blocking antibody (ipilimumab) as combination therapy have significantly improved clinical outcomes for advanced and metastatic melanoma [7,8].

Although checkpoint inhibitors result in long-time survival in some patients with metastatic malignant melanoma, regardless of the *BRAF* mutational status [9], only a group of patients show this favorable response to therapy and these cannot be identified beforehand so far. To understand and decipher the complex nature of melanoma-derived brain metastases, it is essential to establish robust and representative models of melanoma brain metastases that can be generated efficiently. Patient-derived organoid (PDO) models recapitulate the original tumor in terms of tissue architecture and maintain the genetic and histological characteristics of the primary tumor and the intratumoral heterogeneity [10,11]. Therefore, PDO models have emerged as a promising in vitro platform serving diverse research purposes, such as biomarker discovery, personalized medicine, and drug screening, providing enhanced insights into tumor biology and the evaluation of responses to novel therapeutic agents [12]. In this study, we successfully established PDO cultures derived from seven surgical samples of melanoma brain metastases (MBMs), which faithfully retained the genetic and histological characteristics of the primary tumor. Furthermore, the established MBM-PDO cultures were treated in vitro with targeted therapies, *BRAF* and *MEK* inhibitors, to ascertain whether they can accurately predict targeted-therapy responses based on their mutational profile. 

## 3. Material and Methods

### 3.1. Human Specimens

Informed consent was obtained from patients undergoing craniotomies for brain metastases between 2022 and 2023 in the Department of Neurosurgery, University Hospital Bonn. The experiments were approved by the Ethics Committee of the Medical Faculty, University Bonn (#417/17 with amendment from 2020; #169/23). The clinical–pathological characteristics are given in Table 1. The median age at the surgery for brain metastases was 55 years. The primary diagnostic was conducted at the Institute for Neuropathology, University Hospital Bonn. Two patients had brain metastases as the first tumor manifestation, while the other six patients experienced brain metastases at an advanced stage (Table 1).

### 3.2. Human Tumor Collection

Tumor tissues were collected in basis medium ((Advanced DMEM/F12 supplemented with 1× GlutaMax)), 10 mM HEPES solution (Carl Roth, Karlsruhe, Germany), 100 μg/mL Normocin (InvivoGen, San Diego, CA, USA), and 2.5 µg/mL Amphotericin B (Biowest, Nuaillé, France). The tumor samples were manually cut with scissors followed by further mechanical dissociation using gentleMACS™ C Tubes and the gentleMACS™ Octo Dissociator (Miltenyi Biotec, Bergisch Gladbach, Germany). After filtering the tumor samples using a 1000 µm filter, red blood cell lysis was performed using RBC Lysing Buffer Hybri-Max (Sigma-Aldrich, St. Louis, MO, USA). The resultant cell suspensions were then cryopreserved at −80 °C using CryoStor CS10 media (Stemcell Technologies, Vancouver, BC, Canada) until the completion of pathological evaluation and tumor mutational analysis of the primary brain metastasis at the Institute for Neuropathology, University Hospital Bonn. 

### 3.3. MBM-PDO Floating Culture 

The cryopreserved dissociated tumor samples were thawed and digested in an enzyme mix (1 mg/mL collagenase IV (Rockland Immunochemicals, Limerick, PA, USA)), 15 µg/mL DNase (ThermoFisherScientific, Waltham, MA, USA), and 10 µM Y-27632-HCL Rock inhibitor (Biogems, Westlake Village, CA, USA) in basis medium, at 37 °C for 1 h followed by incubation with 3–5 mL TrypLE Express (Gibco, ThermoFisherScientific, Waltham, MA, USA), and filtered through a 70 µm cell strainer (Avantor, Radnor Township, PA, USA). The cells were seeded in ultra-low attachment (ULA) plates (Stemcell Technologies, Vancouver, BC, Canada) in melanoma brain metastases patient-derived organoid (MBM-PDO) culture media (Appendix A) and cultured in a humidified cell culture incubator (37 °C, 5% CO_2_). The MBM-PDO culture media was refreshed every three days. MBM-PDO cultures were observed daily, photographed every four days, and passaged every 3–4 weeks.

### 3.4. Splitting of PDO Cultures 

The MBM-PDO cultures were gently transferred from the ULA plates using a pipette, washed several times in basis medium, centrifuged, and resuspended in 3–5 mL of TrypLE Express (Gibco, ThermoFisher Scientific, Waltham, MA, USA) and incubated at 37 °C for 10 min. After washing the organoids with basis medium supplemented with 10% FCS, the cell pellet was resuspended in MBM-PDO culture medium, seeded in ultra-low attachment (ULA) plates (Stemcell Technologies, Vancouver, BC, Canada), and incubated at 37 °C. 

### 3.5. Embedding of PDO Cultures 

The MBM-PDO cultures were fixed in 4% paraformaldehyde (PFA) overnight at 4 °C, pelleted, and embedded using HistoGel (Richard-Allan Scientific, San Diego, CA, USA). The samples were allowed to cool and solidify at 4 °C for one day and were then embedded in paraffin following standard protocols.

### 3.6. Immunohistochemistry

For immunohistochemistry (IHC) staining, 2 µm thick sections were cut, deparaffinized, and pre-treated according to the standard protocols in the immunohistochemistry laboratory of the Institute of Pathology. Immunohistochemistry was performed on the Medac platform (Melan A: Agilent, clone A103, dilution 1:100; S100:Medac/Cell Marque, clone 4C4.9, dilution 1:2000; HMB45: Agilent, clone HMB45, dilution 1:400; Ki67: Zytomed, mouse anti-human, dilution 1:250, clone K-2; CD8: Agilent; clone C8/144B, dilution 1:50) or on the Ventana platform (CD4: Roche, clone SP35, ready-to-use-antibody). Slides were counterstained with hematoxylin and examined under the microscope (Olympus BX 50) for the evaluation of reactivity.

### 3.7. Mutational Analysis 

The mutational analysis for brain metastases was conducted either by next-generation sequencing (NGS) (Institute of Pathology) or pyrosequencing (Institute of Neuropathology). For primary tumors, DNA from the paraffin-embedded material was extracted using the Maxwell RSC DNA FFPE Kit (Promega, Madison, WI, USA). DNA isolation from the organoids was carried out using the DNeasy Blood and Tissue kit (Qiagen, Hilden, Germany). 

### 3.8. Next-Generation Sequencing (NGS)

DNA was eluted in 120 µL nuclease-free water, and the concentration was determined on a Quantus™ fluorometer using the QuantiFluor^®^ ONE ds DNA System (Promega). Generation of the sequencing library was performed using a QIAseq™ targeted DNA custom panel (Qiagen) with an input of 40 ng DNA. The amplification products were subjected to next-generation sequencing on an Illumina MiSeq sequencer (Illumina, San Diego, CA, USA). The sequencing data were analyzed for genomic variants using the CLC Genomics Workbench/Server 23 (Qiagen Bioinformatics).

### 3.9. Pyrosequencing of BRAF Codon 600

Pyrosequencing was used to determine the sequence at hotspot codon 600 of the *BRAF* gene. A 122 bp fragment of *BRAF*-exon 15 was amplified using following primers *BRAF*-forward 5′-GAAGACCTCACAGTAAAAATAG-3′ and *BRAF*-reverse 5′-Biotin-ATAGCCTCAATTCTTACCATCC-3′. PCR was performed with the Pyromark PCR Kit (Qiagen) with 15 min at 95 °C, followed by 40 cycles of 94 °C, 60 °C, and 72 °C for 30 s each, and finally, 72 °C for 10 min. Single-stranded DNA templates were purified on Streptavidin Sepharose High-Performance beads (GE Healthcare, Chicago, IL, USA) using the PSQ Vacuum Prep Tool and Worktable (Biotage, Uppsala, Sweden). Pyrosequencing was performed using PyroMark^®^ Gold Reagents (Qiagen) on the Pyromark Q24 instrument (Biotage) with the pyrosequencing primer 5′-AGGTGATTTTGGTCTAGCTA-3′. Positive and negative controls were used to compare the results. The pyrograms were analyzed by PyroMark Q24 software Method 003 (Version number 1.0.10, serial number 000019, Biotage) using the allele quantification (AQ) module to determine the percentage of mutant versus wild-type alleles according to percentage relative peak height.

### 3.10. Treatment with BRAF and MEK Inhibitors

To evaluate drug sensitivity, MBM-PDO cultures were seeded into a low attachment 96-well cell culture plate (SARSTEDT AG & CO. KG, Nümbrecht, Germany) with 50 µL of MBM-PDO culture medium in each well. After 24 h of culture, 50 µL of the treatment medium was added to the organoid cultures. These cultures were subjected to a combination therapy of BRAF and MEK inhibitors (dabrafenib (Cayman Chemical, Ann Arbor, MI, USA) and trametinib (MCE MedChemExpress, Monmouth Junction, NJ, USA)), at various concentrations (1 μM, 0.5 μM, 0.25 μM, and 0.125 μM).

### 3.11. Measurement of Intracellular ATP

After 72 h of drug treatment, the intracellular level of ATP was measured by a Cell Titer-Glo 3D assay (Promega, G9682). Briefly, 100 µL of the Cell Titer-Glo reagent was added to each well followed by incubation at room temperature on an orbital shaker for one hour to ensure adequate cell lysis. After incubation, 100 µL samples were transferred to a white Nunc MicroWell 96-Well, Nunclon Delta-Treated, Flat-Bottom Microplate (Thermo Scientific, Waltham, MA, USA). Luminescence measurements were conducted using a SPARK microplate reader (TECAN, Männedorf, Switzerland) with an integration time of 500 ms at the Institute of Experimental Oncology, University Hospital Bonn. The relative viability was calculated as a percentage, normalized to the vehicle control.

## 4. Results

### 4.1. Establishment and Cultivation of MBM-PDOs

To establish PDOs from melanoma brain metastases, we developed a 3D culture protocol that did not rely on tissue extracellular matrices such as Matrigel or collagen (Figure 1A). The PDO cultures were successfully established from MBM samples in seven out of eight cases (Figure 1B), resulting in an overall success rate of 87.5%. We could cultivate metastases from previously untreated as well as from previously treated brain melanoma metastases (Table 1).

The PDO cultures from different patients exhibited a range of distinct morphologies, which were discernible by light microscopy examination of the histological hematoxylin and eosin (H&E)-stained sections (Figure 1C). Some of these cultures displayed a spherical structure with elongated well-defined borders, while others exhibited a rounded structure, or less structured organoids positioned adjacent to each other (Figure 1C). The intra-culture morphological variance of the PDOs was very low.

The growth rate of the MBM-PDOs was visualized 2–3 times per week using a brightfield microscope. A variability in the growth rate was observed among the MBM-PDO cultures. Some cultures expanded exponentially within 10–14 days, after which a growth plateau was reached. Other cultures proliferated at a slower rate (the growth plateau was reached in 20–25 days). Therefore, the MBM-PDOs with a high rate of proliferation required passaging every 10–14 days, whereas the slower-proliferating organoids were re-passaged every 20–25 days. We further evaluated the organoids’ rate of growth by employing Ki-67 staining and distinguished them as having either a low or a high proliferation rate (≤ or >40% of Ki-67 positive nuclei, respectively) (Appendix A). Five of the organoid cultures exhibited a high proliferation rate, while the remaining two organoid cultures displayed a low proliferation rate, and these distinctions were observed across different days.

### 4.2. MBM-PDOs Preserve Key Histological Features of Their Original Tumors

We could cultivate the organoids over a median of eight passages (range four to twelve passages) and the histological features of the MBM-PDOs remained stable over the passages.

To confirm that the PDOs faithfully recapitulated the histomorphology and histological features observed in the original tumors, we conducted a comparative immunohistochemical (IHC) staining. The cultivated MBM-PDOs exhibited positivity for melanoma markers, including S100, Melan A, and HMB45, consistent with their expression in the parental tumor samples (Figure 2).

However, it is noteworthy that the PDO cultures did not preserve the tumor immune microenvironment, resulting in negative stainings for CD4 and CD8 (Figure 2).

Importantly, the PDO cultures displayed stability in their melanoma immunohistochemistry markers, maintaining positive staining for S100 and Melan A even after multiple passages (Figure 3A,B).

### 4.3. MBM-PDOs Recapitulate the Mutational Profiles of Their Original Tumors

Five out of the seven melanoma brain metastases harbored the *BRAF* V600E mutation, and two out of the seven cases had other less frequent mutations (one *NRAS* mutation, one *KIT* mutation). Those two cases were exposed to a panel next-generation sequencing to test for further mutations. Five out of the seven MBM-PDO cultures fully maintained the mutational profile of the parental brain metastasis tissue. One case with *BRAF* wild type lost in vitro the *POLE* mutation demonstrated in the parental tumor (Table 2). In addition, one MBM-PDO culture acquired a *TERT* mutation that was not found in the parental tumor.

### 4.4. MBM-PDOs with BRAF V600E Mutations Show Therapy Response to BRAF and MEK Inhibitors

The MBM-PDO cultures were treated with a combination of dabrafenib (*BRAF* inhibitor) and trametinib (*MEK* inhibitor) at four different concentrations (1 µM + 1 µM, 0.5 µM + 0.5 µM, 0.25 µM + 0.25 µM, 0.125 µM + 0.125 µM) for three days, followed by a cell viability assessment (Figure 4A). The efficacy of the *BRAF* and *MEK* inhibitors on the responding PDO cultures was visually apparent through morphological changes in the organoids (Figure 4C). MBM-PDO cultures with the *BRAF* V600E mutation exhibited good therapy responses to the targeted therapy involving the *BRAF* and *MEK* inhibitors, resulting in marked reductions in cell viability (<50%), whereas the *BRAF* wild-type cultures showed no changes in cell viability (Figure 4B). All four cases treated with *BRAF* and *MEK* inhibitors had a high proliferation rate (Ki-67 > 40%) (Appendix A).

## 5. Discussion

Patient-derived models, such as organoids and tumoroids, are important tools for studying tumor biology and for testing newly developed treatment modalities [13,14,15,16]. Recently, Sun et al. (2023) [17] described organoids derived from primary mucosal melanomas. These organoids retained the histological and molecular features of the primary tumors and could be utilized for assessing drug therapy responses. Currently, limited data about organoid generation from primary skin melanoma are available [18,19]. 

To our knowledge, this is the first study establishing organoids from melanoma brain metastases. We were able to successfully cultivate seven out of eight cases for up to 10 passages. The one case that failed to grow was a brain metastasis relapse, which recurred following surgical resection of a melanoma brain metastasis, followed by stereotactic radiotherapy and immune checkpoint inhibitor therapy. This may be due to the prior therapy and the fact that we tried to cultivate tissue from a relapsed metastasis. The MBM-PDO cultures were stable in culture for several passages, preserving the histological phenotype of the parental tumors throughout the passages, which is in line with the observations of Sun et al. in mucosal melanoma organoids (Sun et al., 2023) [17]. The inter-patient variation in the morphology of the MBM-PDOs could be associated with genetic heterogeneity, differences in the tumor microenvironment, and variations in cellular composition among the patients’ tumors [20]. The MBM-PDO cultures exhibited varying proliferative activities, as revealed by Ki-67 staining. Some MBM-PDO cultures were highly proliferative (>40% positive nuclei), while others were lowly proliferative (<40%). The proliferation rate was concordant between the MBM-PDOs and their parental tumors. The absence of a tumor immune microenvironment in the generated MBM-PDO culture limits the model’s ability to accurately recapitulate tumor–immune interactions. To address this limitation, several strategies can be employed in the future such as autologous or allogeneic immune cells and organoid co-culture systems [21]. Additionally, the inclusion of specific cytokines and growth factors, such as interleukin-2 (IL-2), can enhance the viability and proliferation of immune cells within the organoids, thereby improving the recapitulation of the in vivo microenvironment [22]. We assessed the mutational status of the parental brain metastasis tissue and the MBM-PDO cultures and observed that *BRAF* mutations were conserved in culture. Five out of seven MBM-PDO cultures (71%) fully recapitulated the mutational profile of their parental tumors. Notably, both MBM-PDO cultures from *BRAF* wild-type melanomas also had *TERT* promoter mutations. *TERT* mutations are common in melanoma (69%) and are associated with a poor prognosis [23]. The most frequent mutation in primary brain tumors as well as in metastasis is the *C250T* mutation, corresponding to the mutation observed in our two cases. Interestingly, Blanco-Garcia et al. [23] also noticed that the *C250T TERT* mutation was often associated with *NRAS* mutations, which was detected in one of our two cases with a *TERT* mutation. To demonstrate the close molecular similarity between the MBM-PDOs and their parental tissue, transcriptomic profiles from the MBM-PDOs as well as their parental tissue could be analyzed via bulk RNA sequencing. Melanoma brain metastases are the primary cause of death in 60–70% of melanoma cases [24]. Since the introduction of tyrosine kinase inhibitors and the MAPK inhibitors for the treatment of metastatic melanomas, the overall survival of patients increased dramatically. Two clinical studies, COMBI-d and COMBI-v, conducted on patients with metastasized melanoma and *BRAF* V600E and *V600K* mutations, showed a 5-year overall survival rate of 34% and a median overall survival time of 25.9 months with dabrafenib plus trametinib treatment [25]. For melanoma brain metastases with *BRAF V600* mutations, a combination therapy with dabrafenib and trametinib is effective; however, the responses are less durable than those of extracranial metastases with the same mutations [26]. The COMBI-r study reported a 10.8 months overall survival for patients with melanoma brain metastases with *BRAF V600E* or *V600K* mutations treated with dabrafenib/trametinib as the first-line therapy [27]. We aimed to test the feasibility of utilizing MBM-PDO cultures as an in vitro platform to identify the efficacy of targeted therapy. To that end, we treated four cultures with a combination of dabrafenib/trametinib. As expected and described by Sun et al., the treatment response correlated with the mutational status of the tumors [17]. Two MBM-PDO cultures harboring *BRAF* V600E mutations had a very good response to the combination therapy, as assessed by cell viability, while the *BRAF* wild-type cultures were insensitive. Interestingly, the MBM-PDOs harboring an *NRAS* mutation were also insensitive to therapy. Since we treated the PDOs with *MEK* inhibitors, besides *BRAF* inhibitors, we expected a therapy response, even if it was not as pronounced as for PDOs with *BRAF* mutations. This underlines the individualized patients’ response to targeted therapy.

In conclusion, we successfully established patient-derived organoids from melanoma brain metastases, which faithfully recapitulated the histological and mutational characteristics in culture, over passages. In vitro drug testing demonstrated the capacity of the MBM-PDOs to reveal targeted-therapy susceptibilities, which highlights their great potential in preclinical research, drug discovery, and personalized medicine.

## Figures and Tables

**Figure 1 pharmaceutics-16-01042-f001:**
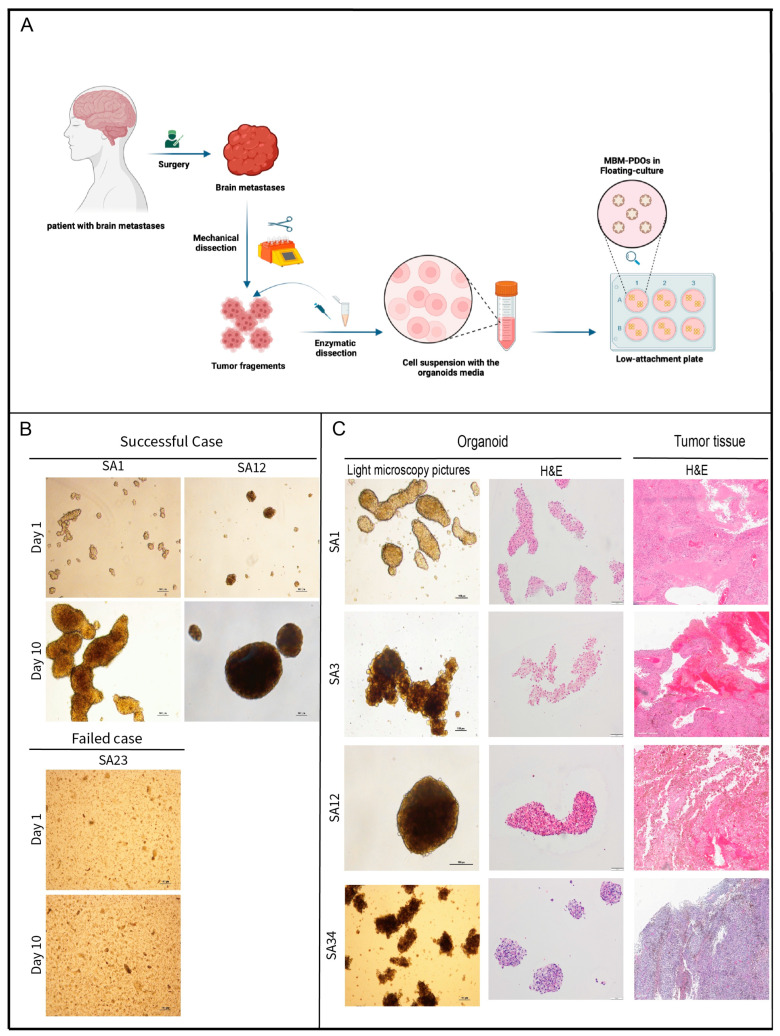
(**A**) Graphical representation for the MBM-PDOs’ generation workflow. Tumor samples are dissociated into single-cell suspensions and grown in low-attachment plates (created with BioRender.com). (**B**) Bright-field images of PDOs after 1 day and 10 days of culture establishment (scale bar 100 µm). (**C**) Bright-field images (5× magnification) and hematoxylin and eosin (H&E) staining (10× magnification) showing different phenotypes of MBM-PDOs.

**Figure 2 pharmaceutics-16-01042-f002:**
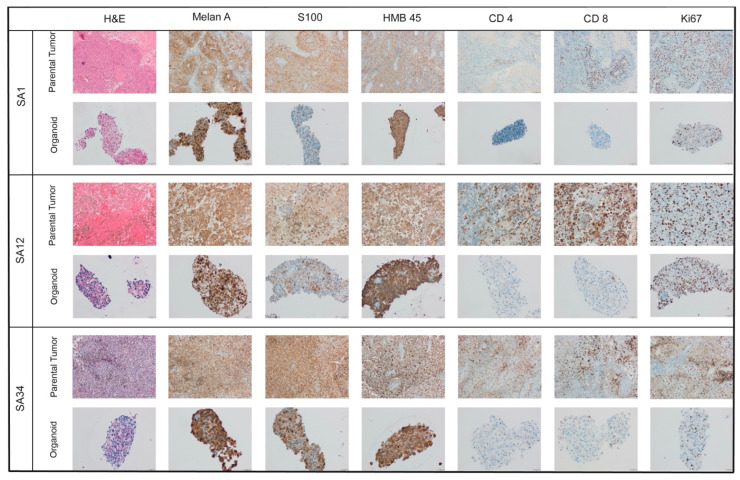
Representative images of hematoxylin and eosin (H&E) staining and immunohistochemical staining (for Melan A, S100, HMB45, CD4, CD8, and Ki67) of the parental tumors and their MBM-PDO cultures (10× and 20× magnification).

**Figure 3 pharmaceutics-16-01042-f003:**
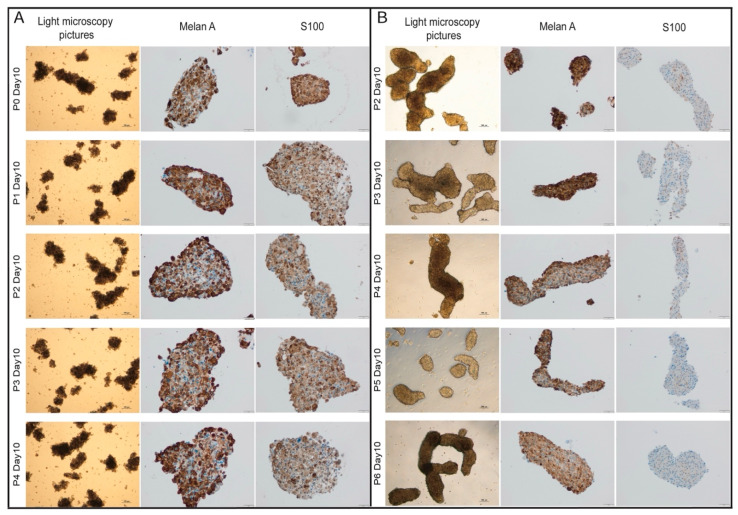
Bright-field images (5× magnification) and representative images of immunohistochemical stainings for Melan A and S100 staining (10× magnification) of different passages showing the stability of organoid-morphology and immunohistochemical profile. (**A**) SA34 MBM-PDO culture. (**B**) SA1 MBM-PDO culture. P, passage.

**Figure 4 pharmaceutics-16-01042-f004:**
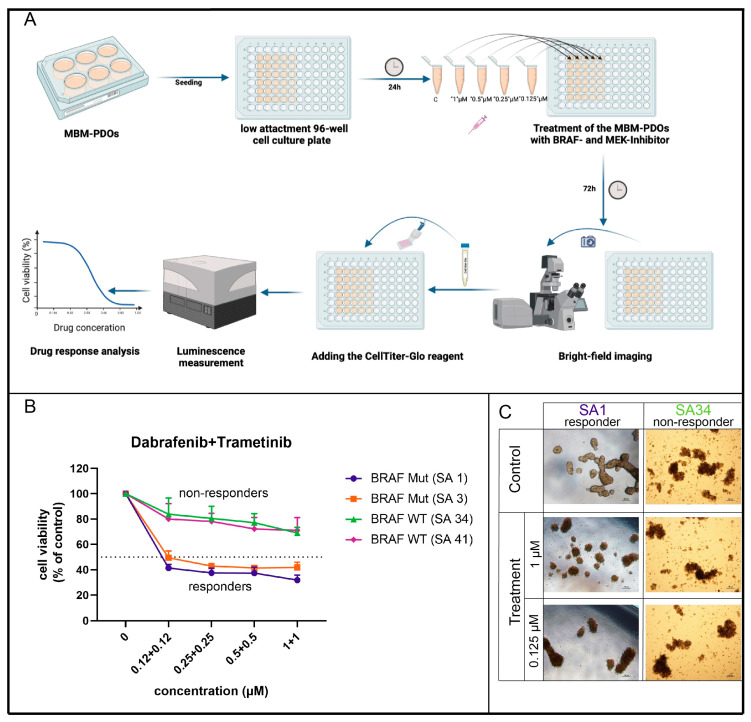
(**A**) Graphical abstract for the in vitro drug sensitivity assay (created with Biorender.com). (**B**) A dose–response graph depicting the cell viability of MBM-PDO cultures treated with different concentrations of dabrafenib and trametinib for 72 h. Cell viability, assessed using the CellTiter-Glo assay (Promega), was normalized to the vehicle control. Each condition was tested in technical and biological triplicates. Data are presented as mean ± SEM. (**C**) Representative brightfield microscopy images of MBM-PDO cultures taken after 72 h of treatment with 1 µM and 0.125 µM dabrafenib and trametinib, as well as vehicle control (top row) (scale bar 100 µm).

**Table 1 pharmaceutics-16-01042-t001:** Clinico-pathological characteristics of the patients included in this study.

Characteristics			Number (%)
Sex		Male	5 (62.5%)
		Female	3 (37.5%)
Localization		Frontal	4 (50%)
		Parietal	3 (37.5%)
		Occipital	1 (12.5%)
Number of brain metastases		Singular	6 (75%)
		Multiple	2 (25%)
Extracranial metastases		0	5 (62.5%)
		Pulmonary	1 (12.5%)
		Osseous	2 (25%)
TNM *	T	pTx (first diagnosis)	2 (25%)
		pT1	0 (0%)
		pT2	1 (12.5%)
		pT3	4 (50%)
		pT4	1 (12.5%)
	N	pN0	5 (62.5%)
		pN1	2 (25%)
		pN2	1 (12.5%)
	M	pM1	8 (100%)
Mutation		BRAF V600E	5 (62.5%)
		BRAF wild type	3 (37.5%)
Therapy	Pre-neurosurgical resection	Stereotactic radiotherapy	3 (37.5%)
		Interferon therapy	2 (25%)
		Combination immunotherapy	3 (37.5%)
	Post-neurosurgical resection	BRAF and MEK inhibitor (tafinlar, mekinist)	2 (25%)
		Combination immunotherapy (nivolumab, ipilimumab)	3 (37.5%)

* TNM classification: T (tumor) represents the size and extent of the primary tumor; N (node) represents the involvement of regional lymph nodes; M (metastasis) represents the presence of distant metastasis.

**Table 2 pharmaceutics-16-01042-t002:** Mutational analysis of the primary tumors and paired PDO.

Case	Primary Tumor	Organoid
SA1	*BRAF V600 E*	*BRAF V600 E*
SA3	*BRAF V600 E*	*BRAF V600 E*
SA12	*BRAF V600 E*	*BRAF V600 E*
SA17	*BRAF V600 E*	*BRAF V600 E*
SA20	*BRAF V600 E*	*BRAF V600 E*
SA34	*BRAF V600 WT*	*BRAF V600 WT*
	*NRAS p.Q61K*	*NRAS p.Q61K*
		*TERT c.146C>T*
SA41	*BRAF V600 WT*	*BRAF V600 WT*
	*KIT p.L576P*	*KIT p.L576P.*
	*TERT c.146C>T*	*TERT c.146C>T*
	*TERT c.125_124delinsTT*	*TERT c.125_124delinsTT*
	*POLE c.1360-1>A*	

## Data Availability

The datasets analyzed during the current study are available from the corresponding author on reasonable request.

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
