# Peer review of "Melanoma Brain Metastases Patient-Derived Organoids: An In Vitro Platform for Drug Screening"

_pharmaceutics, 2024, doi:10.3390/pharmaceutics16081042_

Round 1

Reviewer 1 Report

Comments and Suggestions for Authors

The manuscript entitled "Melanoma Brain Metastases Patient-Derived Organoids: an in Vitro Platform for Drug Screening" is well written and contains valuable results that can be considered for publication. However, a few clarifications should be addressed by the authors.

1- The choice of the passage period (3-4 weeks) should be explained to enable the readers to know the reason for selecting this period. In my opinion, this period is too long. Did the authors try other culture media to see if this can enhance the growth of PDO-Cultures?

2- In the discussion section, I was expecting the authors to explain why PDO-cultures have different morphologies amongst patients.

Author Response

Dear Reviewer,

Thank you for your valuable feedback on our manuscript "Melanoma Brain Metastases Patient-Derived Organoids: an in Vitro Platform for Drug Screening." We appreciate your thorough review and constructive comments. We tried to response all the questions raised by the reviewer and modified our manuscript accordingly.

Comment 1: “The choice of the passage period (3-4 weeks) should be explained to enable the readers to know the reason for selecting this period. In my opinion, this period is too long. Did the authors try other culture media to see if this can enhance the growth of PDO-Cultures?”

The choice of the passage period for our MBM-PDO cultures was based on careful monitoring of the organoid cultures every week, 2-3 times, using a light microscope to visually evaluate their growth. We observed that fast-growing PDO cultures ceased to show significant growth and began to lose their organoid structure after 10-14 days, while slow-growing PDO cultures exhibited the same behavior after 20-25 days. To understand this variability, we investigated the Ki67 index of the PDOs and their corresponding parental tumors. We found that fast-growing PDO cultures had a high Ki67 index (>40%), mirroring their parental tumors, whereas slow-growing PDO cultures had a lower Ki67 index (<40%), also reflecting their parental tumors (lines 211- 215, and supplementary table 2). This correlation informed our decision to select different passage periods, which we will elaborate on in the first results section of our manuscript. There is indeed limited information available on melanoma organoid cultures. We used a generic organoid media that has been described by previous publications (1) for culturing organoids from other tumor entities and did not test other media to see if there would be differences in growth or morphology. However, we did explore different culture methods to optimize MBM-PDO growth, such as dome cultures and air-liquid interface (ALI) cultures. The best growth results were achieved with these methods, as detailed in our manuscript.

Comment 2:” In the discussion section, I was expecting the authors to explain why PDO-cultures have different morphologies amongst patients“

We have now explained and added information regarding the varying morphologies of PDO cultures among patients (291-293).

We hope these revisions address your concerns and enhance the value of the manuscript. Thank you again for your insightful comments.

References

  1. Neal JT, Li X, Zhu J, Giangarra V, Grzeskowiak CL, Ju J, et al. Organoid Modeling of the Tumor Immune Microenvironment. Cell. 2018 Dec 13;175(7):1972-1988.e16.

Best regards,

Saif-Eldin Abedellatif

Prof. Dr. Marieta Ioana Toma

Reviewer 2 Report

Comments and Suggestions for Authors

The work by Abedellatif describes an interesting strategy to generate melanoma-derived brain tumor organoids to be used as a platform for drug discovery. The work is extremely relevant, as those deadly tumors need to be further studied and those patients tend to be under-represented in clinical trials.

The methods described are sufficiently detailed and the data appears to be sound, supporting their conclusions.

This reviewer would prefer that the authors expanded the characterization of their models by, for example, sparing half of the primary material for RNA extraction, cultivating the other half in their model and comparing RNAseq profiles between the two types of samples to test the fidelity of the gene expression landscapes more broadly in their model. While the authors do monitor the preservation of specific BRAF, NRAS and TERT mutations of interest, that is a very limited view on what can prove to be an even more powerful model.

Thus, it would be desirable that the authors addressed the comment above in the discussion, as perhaps a future direction for follow-up work.

Author Response

Dear Reviewer,

Thank you for your valuable feedback on our manuscript "Melanoma Brain Metastases Patient-Derived Organoids: an in Vitro Platform for Drug Screening." We appreciate your thorough review and constructive comments and modified the discussion part accordingly.

Comment: We have now addressed your comment in the discussion section, proposing that RNA from the PDO cultures can be sequenced and compared with RNA sequencing from the primary tumor to demonstrate the close molecular similarity and closeness of the gene expression profiles between the PDO cultures and the primary tumor (lines 315-318).

We hope these revisions address your concerns and enhance the manuscript. Thank you again for your insightful comments.

Best regards,

Saif-Eldin Abedellatif

Prof. Dr. Marieta Ioana Toma

Reviewer 3 Report

Comments and Suggestions for Authors

The authors designed the study for preparing Melanoma Brain Metastases Patient-Derived Organoids: An In Vitro Platform for Drug Screening. The work was well-designed and showed promising outcomes. However, there are some of minor concerns with the manuscript that need to be addressed before it can be accepted.

This study would be among the first works to establishing organoids from melanoma brain metastases which can be used for the next drug screen and responses evaluation.

Minor Comments:

1.    The introduction is well-written, and the aim is clearly indicated. Additionally, the methodology is detailed and well-explained, which is appreciated.

2.    Please define the abbreviation of TNM in Table 1: T describes the size of the tumor and any spread of cancer into nearby tissue; N describes spread of cancer to nearby lymph nodes; and M describes metastasis (spread of cancer to other parts of the body).

3.     In line 230, the authors mentioned that the PDO cultures did not preserve the tumor immune microenvironment, resulting in negative staining for CD4 and CD8. It would be valuable to assess immune cell activity in the TME, as they are a crucial component of patient cancer treatment. Therefore, the authors could consider co-culturing the PDOs with autologous or allogeneic immune cells, such as T cells (including CD4+ and CD8+), dendritic cells, or other immune cells relevant to the TME. Alternatively, they could supplement the PDO culture medium with cytokines and growth factors that support the survival and function of immune cells, such as IL-2, IL-7, IL-15, and GM-CSF.

4.    The authors have only checked the cell toxicity of the BRAF and MEK inhibitors in combination. It might be better to also test them separately for comparison.

5.    Besides viability, consider assessing additional endpoints experiments such as the expression of proteins involved in the MAPK/ERK pathway and apoptosis, such as pERK, cleaved PARP, caspase, BCL-2, and BAX activity through western blotting, and flow cytometry to analyze cell cycle progression after treatment would provide a better understanding of the organoid cell response to the treatment.

Author Response

Dear Reviewer,

Thank you for your valuable feedback on our manuscript "Melanoma Brain Metastases Patient-Derived Organoids: an in Vitro Platform for Drug Screening." We appreciate your thorough review and constructive comments.We modified our manuscript accordingly and tried to answer all the questions.

Comment 2: “Please define the abbreviation of TNM in Table 1”

We have now described the abbreviation of the TNM classification in Table 1.

Comment 3: “In line 230, the authors mentioned that the PDO cultures did not preserve the tumor immune microenvironment, resulting in negative staining for CD4 and CD8. It would be valuable to assess immune cell activity in the TME, as they are a crucial component of patient cancer treatment. Therefore, the authors could consider co-culturing the PDOs with autologous or allogeneic immune cells, such as T cells (including CD4+ and CD8+), dendritic cells, or other immune cells relevant to the TME. Alternatively, they could supplement the PDO culture medium with cytokines and growth factors that support the survival and function of immune cells, such as IL-2, IL-7, IL-15, and GM-CSF“

This is indeed a limitation of the current model. We have addressed these limitations in the discussion (lines 279-304).

Comment 4: “The authors have only checked the cell toxicity of the BRAF and MEK inhibitors in combination. It might be better to also test them separately for comparison“

We checked the cell toxicity of the BRAF and MEK inhibitors as a combination therapy to mimic the current patient guidelines for melanoma patients harboring BRAF-mutated tumors, rather than testing the substances separately. Since we were working with limited material, we had to choose carefully the substances and conditions we tested and also repeat the experiments three times.

Comment 5: “Besides viability, consider assessing additional endpoints experiments such as the expression of proteins involved in the MAPK/ERK pathway and apoptosis, such as pERK, cleaved PARP, caspase, BCL-2, and BAX activity through western blotting, and flow cytometry to analyze cell cycle progression after treatment would provide a better understanding of the organoid cell response to the treatment “

Although this would have been interestingly to assess, there is a technical limitation to performing western blots which is due to the scarcity of the PDO culture material (usually, a lot of cells (in the million range) are required to get a good protein concentration for performing western blots). We plan to continuate this project, refine the organoid dissociation to single cell, since the PDO cells tend to form cell clumps shortly after dissociation since the PDO cells tend to form cell clumps shortly after dissociation, and also perform further analysis, like cell cycle and apoptosis.

We hope these revisions address your concerns. Thank you again for your insightful comments.

Best regards,

Saif-Eldin Abedellatif

Prof. Dr. Marieta Ioana Toma